# Towards an inclusive nature conservation initiative: Preliminary assessment of stakeholders' representations about the Makay region, Madagascar

Céline Fromont[1¤]*, Julien Blanco[1], Christian Culas[2], Emmanuel Pannier[3], Mireille Razafindrakoto[4], François Roubaud[4], Stéphanie M. Carrière[1]*

1 IRD-Montpellier, UMR Knowledge, Environment and Societies (SENS), IRD, CIRAD, Université Paul-Valéry, Université de Montpellier, Montpellier, France, 2 CNRS, UMR Actors, Resources and Territories in Development (Art-Dev), Université Paul-Valéry, Montpellier, France, 3 IRD, UMR Local Heritage, Environment and Globalization (PALOC), Paris, France, 4 IRD, UMR LEDa, DIAL, Université Paris-Dauphine, PSL Research University, Paris, France

¤ Current address: CEFE, Université de Montpellier, CNRS, EPHE-PSL University, IRD, Montpellier, France
* celine.fromont@cefe.cnrs.fr (CF); stephanie.carriere@ird.fr (SMC)

**Data Availability Statement:** The data collected during the research have been archived in the online, publicly available database DataSuds of the French national research Institute for Sustainable

## Abstract

The existence of multiple perspectives and representations of different stakeholders poses critical challenges to conservation initiatives worldwide. Thus, to foster more just and sustainable agendas in protected areas (PAs), this diversity of perspectives must be better understood, acknowledged, and tackled. In this article, we aimed to initiate this understanding for the Makay region in Madagascar, a poorly-known region where a 'New Protected Area' has been gazetted. In combining mental models and social representation theory, we explored different stakeholders' perspectives about the Makay social-ecological system, and how differences in stakeholders' viewpoints could challenge the success of an inclusive, just, and sustainable conservation program. We conducted semi-structured interviews with 32 respondents having different expertise on the Makay. During interviews, respondents were guided towards the elicitation of their individual cognitive map (ICM) of the Makay social-ecological system. ICMs were then analyzed in combining quantitative and qualitative. Respondents described the Makay through a total of 162 components, including 51 components that constituted the central zone of the Makay's representation. In particular, respondents pointed to insecurity issues caused by zebu thieves, as well as to environmental challenges relative to anthropogenic fires and hunting. On the contrary, they considered mining activities and timber harvesting as more peripheral problems. Through a multivariate clustering analysis, we discriminated two clusters of respondents with contrasting visions about the Makay, ecocentric vs. social-ecological, which was largely influenced by respondents' background. In comparing the two clusters' representations, we found that they had dissimilar diagnoses about key socio-environmental challenges in the Makay and how to address them. This ambiguity in respondents' viewpoints stresses the need to increase research efforts in the Makay region to fill current knowledge gaps about this poorly known

Development (IRD) (https://dataverse.ird.fr):
https://doi.org/10.23708/RWDMN9.

**Funding:** SMC and CC received funding from the
"Fondation Maison des Sciences de l'Homme" for
the MakayMada project of which this study is part.
The funders had no role in study design, data
collection and analysis, decision to publish, or
preparation of the manuscript.

**Competing interests:** The authors have declared
that no competing interests exist.

social-ecological system, and to foster social learning between stakeholders concerned by
the Makay new PA.

## 1. Introduction

Protected areas (PAs) are the main tool for promoting nature conservation worldwide, and are
increasingly understood as complex social-ecological systems (SESs) that must tackle both bio-
diversity and local development issues [1, 2]. For instance, in Madagascar, 'New Protected
Areas' aim at tackling the joint challenge of reconciling human development and nature con-
servation but were shown to have limited effectiveness in reducing deforestation and other
threats on biodiversity, mainly because of their rapid establishment processes and the difficulty
to meet multiple objectives in a context of financial and human resource scarcity [3, 4].
Another critical challenge to the social and ecological success of PAs stands in the innumerable
and sometimes diverging knowledge, beliefs and values of the many actors who either inhabit
them, beneficiate from them, and/or manage them. This diversity does not only generate dif-
ferent ways of understanding, formulating and addressing nature conservation problems, but
also underpins tensions and conflicts between people, undermining their ability to reach a
consensus on PA management strategies and solve related collective problems [5–9].

The existence of multiple possible interpretations of a situation relates to one kind of uncer-
tainty that has been referred to as '*ambiguity*' [10–12]. Contrary to '*epistemic uncertainty*',
ambiguity is not strictly caused by a lack of knowledge on a situation but rather by the fact that
there are different sensible, valid and legitimate ways of understanding the situation among
people. In other words, ambiguity is a key feature of wicked problems that, by definition, have
no definitive formulation and a plethora of plausible solutions [13]. As part of such wicked
problems, nature conservation challenges therefore require to pay careful attention to view-
point diversity, and to engage with ambiguity instead of seeking to reduce it [14–17]. In this
perspective, a first key step is to characterize viewpoint diversity, in which might help reveal
what could undermine people's ability to sustain collective actions towards more effective
nature conservation strategies [18, 19].

This study aimed at this objective in the Makay region in Madagascar, where the creation of
a terrestrial PA is currently an on-going process. The Makay region is a particularly isolated
mountainous area that has been officially 'discovered' in 2001 by the Western world [20]. As a
consequence, scientific research in this region is scarce: the Scopus database counts 10 docu-
ments on the Makay, most of which reporting findings about new species (S1 Table). Notwith-
standing the poor understanding of this region–its unique ecosystems but also its local people
and their relationships with nature–it is already promoted as a unique place for its outstanding
biodiversity and preserved ecosystems. This promotion has led to a nurturing nature conserva-
tion project which took shape in 2017 with the creation of a New Protected Area through a
temporary decree. While this new PA aims to promote the sustainable management of the
region through both development and conservation actions, one might assume that this gen-
eral objective could be jeopardized by (i) the lack of scientific basis to orient management
choices and strategies, which is a fertile ground for the emergence of (ii) tensions and conflicts
between different stakeholders' representations, values and knowledge.

In this context, the aim of this paper is to provide a first general understanding of the
Makay SES in a multi-stakeholder perspective, and to clarify the different points of view that
coexist among stakeholders, especially about nature and people interactions. To do so, we
relied on the combination of two complementary theories and tools: mental models on the

one hand, and social representations on the other hand. The former allowed us to assess how stakeholders perceived the functioning of this SES as a complex system where multiple bio-physical components interact with multiple social components. While this approach was shown promising to help integrate different sources of knowledge [21], co-construct a shared vision of an SES [22], or characterize ambiguity between different actors [23], it remains poorly used in the field of conservation biology [24]. The latter was used to discriminate the elements that were central in the Makay representations from more peripheral elements, which ultimately allowed us to identify different types of ambiguity between people's view-points. On the basis of our results, we draw three key recommendations that could help foster a just and sustainable conservation program for the Makay region.

## 2. Study site

### 2.1. Presentation of the Makay region

The Makay mountain range is located in the southwestern part of Madagascar in a particularly isolated area (Fig 1). It is a vast detritic massif (4000 km$^2$) characterized by long, deep canyons created by the erosion of an ancient mountain range [25]. Its unique geomorphology allows a great diversity of micro-habitats, ecosystems and vegetation types to coexist in the same geo-graphical area. Indeed, we found specific herbaceous habitats on the arid top of the plateaus and woody forests in the humid bottom of the canyons, whose diversity of width and orienta-tion provides varied conditions of humidity and temperature specific to each canyon. This diversity of micro-habitats is occupied by very contrasting series of vegetation, typical of both the dry forests of the east of the island and the humid forests of the west, in the same geograph-ical location: gallery forests at the bottom of the canyons, very marked vegetation gradient on the cliffs, xerophilous vegetation and patches of dry forests on the plateaus. Thus, scientific expeditions have shown that the Makay massif is home to a remarkable biodiversity, and have led to the discovery of a hundred endemic species of Makay, the region or the country. Archae-ological remains, including the first rock paintings in Madagascar discovered in the Makay, prove that the region has long been inhabited. Nowadays, there are no permanent settlements within the Makay itself, but numerous villages of less than 500 inhabitants, sparsely populated and dispersed, are established around the area [20]. The extreme isolation of Makay compli-cates the economic exchanges of these people with the rest of the country. Local people are farmers and zebu raisers, and use and manage the resources of the Makay, especially for access to pasture, fresh water, and the harvesting of tubers or medicinal plants.

### 2.2. Challenges and potential threats

With a particularly high concentration of endemic species associated with high rates of defor-estation, Madagascar is one of the world's hottest hotspots [26–28]. At the same time, Mada-gascar faces major and urgent economic and social development challenges [28, 29]. The country is ranked 164 in the United Nations Development Programme's (UNDP) Human Development Index (HDI) [30], and did not meet any of the Millennium Development Goals (MDG) between 2000 and 2015 [31]. In rural areas, most people depend on the collection of natural resources in forest areas [28]. Coupled with state instability and weakness, the coun-try's socioeconomic situation is partly responsible for the manifold causes of deforestation, which include slash-and-burn agriculture, illegal logging of precious woods, uncontrolled fires and mining [29, 32]. This socioeconomic situation is particularly noteworthy in remote rural areas, like the Makay region, where state development policies have little effect. Thus, expedi-tions to the Makay have revealed, but not scientifically substantiated, that the Makay is subject to threats of varying degrees, such as destruction of the forest cover by bush fires, logging,

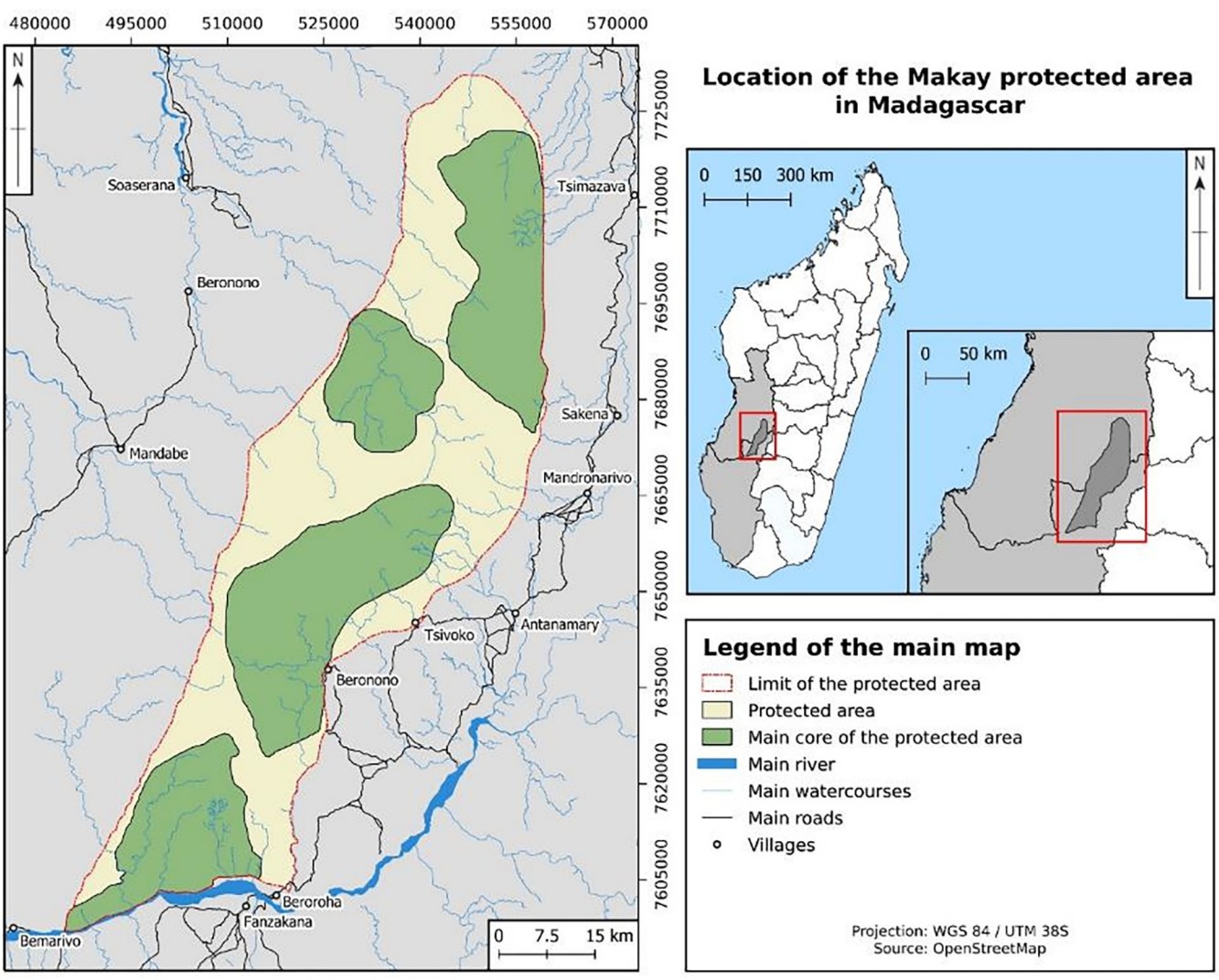

**Fig 1. Location of the Makay region, Madagascar.** The map was built by Auréa Pottier (engineer at IRD), under QGIS, using OpenStreetMap data.

erosion, destructive honey harvesting, and poaching, which poses a direct threat to the species. Thus, the study of these potential threats to the remarkable biodiversity of the Makay, and of the interdependence between the massif and the local populations, appears to be a determining factor in reconciling conservation and development in this region. To face these challenges, NGOs work on the creation of a new protected area to preserve this region.

## 3. Material and methods

### 3.1 Conceptual background

In order to explore the multiple representations of the Makay SES at individual and collective levels, we combined in this article two complementary theories and associated methods.

Firstly, we mobilized a mental-model approach for its capacity to allow both qualitative and quantitative analyses of people's representations while being suited to situations where data are scarce and poorly reliable [33]. Mental models has been defined as internal, cognitive models that underlie how people perceive and understand the cause and effect relationships that

drive their environment [34]. As internal representations, people's mental models are not accessible to researchers and can only be approximated through diverse, direct or indirect, elicitation procedures [35]. Among these, cognitive mapping refers to a widespread family of elicitation procedures that consists in representing people's mental models as diagrams called cognitive, or causal, maps composed with nodes–the elements that respondents used to describe their reality–and edges–the relations between these elements as viewed by respondents [33, 36]. Cognitive maps offer a systemic representation of people's representations, and are acclaimed from their ability to allow in-depth qualitative analyses [37, 38] as well as quantitative analyses based on graph theory [33, 39]. However, cognitive maps are insufficient to explore how an individual acquires a given mental model [6]. Furthermore, the consequences of differences in people's cognitive maps are difficult to grasp with the sole mental-model theory.

To address these shortcomings, we secondly mobilized the social representation theory and, more precisely, the structural approach of social representations (for a review, see [40]. Social representations are both a cognitive system at the social level that allows individuals and groups to construct a coherent vision of reality, and the outcome of mental activity modulated by the social context and interactions with other people [41]. According to the structural approach, the content of a social representation of an object or a situation is composed with a nucleus and a peripheral zone: the former structures and stabilizes the representation, containing elements that are shared within a social group; the latter allows for adaptation, flexibility and individual variation, containing less shared elements [42]. These are the peripheral elements that are likely to reflect differences in people's viewpoints regarding a social object, i.e. ambiguity sensu [10]. More precisely, researchers generally identify four zones of the representation (i.e. central core zone, contrasting elements' zone, first periphery, and second periphery) on the basis of the frequency of citation of each element and of the importance respondents assign to it [43, 44]. Each of these zones having specific functions and meanings, this approach ultimately allows to understand the causes and consequences of the coexistence of a plurality of visions of a given social object within a human society.

## 3.2. Inclusivity in global research

The French national research Institute for Sustainable Development (IRD) does not deliver ethics approval for research involving human participants. We followed the recommendations of the European General Data Protection Regulation (GDPR). In particular, before interviews, respondents were informed of the purpose and aim of the research and of the envisioned use of the data collected (i.e. for research purpose only), which allowed us to obtain their Free, Prior and Informed Consent (FPIC). FPIC were collected orally as most interviews were conducted remotely. During interviews, we only collected personal data that was strictly necessary to the research, i.e., respondents' name and professional activity. Finally, respondents' anonymity was ensured by pseudo-anonymizing data (e.g. transcripts and mental models) and by deleting audio records. Only the first author, who was responsible for data analyses, had access to the data de-anonymization key. Because this study did not involve local people, it did not fall into the scope of ethical codes or legislations of Nagoya protocol that aim to protect traditional knowledge. Additional information regarding the ethical, cultural, and scientific considerations specific to inclusivity in global research is included in the Supporting Information (S1 Checklist).

## 3.3. Data collection

The overall method we followed is summarized in Fig 2 and further details in Supporting Information. In order to obtain a comprehensive view of the Makay region and context-

**Step 1** *Eliciting individual representations*
*about the Makay as cognivite maps*

Semi-structured interviews

⇒ Individual cognitive maps (ICMs) of the Makay SES

**Step 2** *Condensing the ICMs*
*and homogenizing cited components*

Qualitative standardization

⇒ Comparable ICMs of the Makay SES with different levels of details

**Step 3** *Analyzing the content and structure*
*of Makay social representations*

Centrality-frequency criteria

⇒ Identification of the four zones of the representation
⇒ ICM aggregation to elaborate the social cognitive map (SCM)

**Step 4** *Characterizing ambiguity*
*in people's representations*

Quantitative & qualitative analyses

⇒ Groups of individuals based on their representation of the Makay
⇒ Ambiguity in people's representations at individual and group levels

**Fig 2.** Overview of the four-step method: (i) elicitation of individual cognitive maps (ICMs), (ii) ICM condensation, (iii) analysis of the social representation and production of a social cognitive map (SCM) of the SES, and (iv) analysis of ambiguity at individual and group level through multivariate analyses and production of group cognitive maps.

specific social-ecological challenges, we targeted stakeholders who were prone to participate in–and/or influence–the management of the Makay's protected area, either because they worked in the region or had a level of expertise about it (either scientific or experiential), and who have been to the Makay at least once. Yet, because of logistical constraints, we were not able to go to the Makay itself, which prevented us from including local people in this study, which we yet consider as a priority research perspective.

As a result, we interviewed a total of 32 respondents, including (i) 12 researchers with different academic backgrounds and fields of expertise, (ii) 9 members of local environmental organizations working in the Makay, including members of the NGO carrying the project of creation of the protected area and expected to be the future manager of this protected area (iii) 6 researchers involved in an interdisciplinary research program evaluating Makay socio-economic and environmental factors, and (iv) 5 representatives of Makay tour operators. Respondents were identified through a purposive sampling strategy: each interview contained questions to identify additional relevant people to meet. Sample size was determined by a data saturation technique: we stopped interviewing more people when no new information or additional respondents was added (S1 Fig in S1 Appendix).

To elicitate respondents' mental models of the Makay SES, we developed a semi-structured interview procedure that was inspired from the ARDI procedure, a well-known method that aims to untap the Actors, Resources, Dynamics, and Interactions of an SES as a cognitive map [22]. Each interview was conducted as follow. First, we introduced the concept of cognitive maps to the respondent and, with the help of an example not related to the Makay, we familiarized them with the Mental Modeler software that allows to create cognitive maps thanks to a user-friendly interface and that we chose to use to facilitate the direct elicitation of respondents' cognitive map [45]. We then reminded the respondent of the study area (the Makay massif and the eastern villages), and explained the purpose and organization of the direct elicitation procedure. Second, through successive inductive questions, the respondent was asked to freely cite the different elements they associated with the Makay SES, including its (i) biophysical characteristics (e.g. living and non-living entities, natural and human-modified landscape features) and associated ecosystem services, (ii) main stakeholders, i.e. individuals or groups of people who play a major role in or benefit from the SES, and (iii) main dynamics, processes and drivers affecting the SES. All these items were considered 'components' of the SES, i.e. as nodes in the cognitive maps. Finally, the respondent was asked to identify, if relevant to them, positive (+1) and negative causal links (-1) between components, which were considered 'interactions' within the SES, i.e. as edges in the cognitive maps. In order to limit framing bias (i.e. when the researcher unintentionally leads respondents to add a component or an interaction from his/her own beliefs), the researcher strictly followed the indications provided by the respondent to draw the map on Mental Modeler, fostering continuous feedbacks between the map under progress and the conversation with respondents. All respondents were eventually asked to confirm that the resulting map satisfactorily depicted their vision of the Makay (see S2 Fig in S1 Appendix for an example). Interviews were conducted face to face or by videoconference, lasting from one to four hours.

## 3.4. Data analysis

Prior to formal analyses, we followed a qualitative condensing procedure that aimed at making individual cognitive maps (ICMs) comparable without substantially changing their meaning (Fig 2, S1 Appendix). A first level of standardization led to the identification of 162 unique components, which is considered too high for the interpretation of cognitive maps [33], so we further grouped them into 32 types (S1 Table) and six larger categories (i.e. biophysical and

social components and processes, positive and negative human interventions, and ecosystem services and disservices).We then imported condensed ICMs as adjacency matrices into R statistical software [46]. Matrix additions first allowed us to aggregate all the 32 ICMs into a social cognitive map (SCM), offering a synthetic vision of the Makay SES as held by our respondents. Second, matrix additions allowed us to focus on certain environmental challenges (such as fires) and to highlight in a comprehensive way how these challenges were perceived by our different respondents.

To analyze the diversity of representations among the respondents about the Makay SES, we mixed quantitative and qualitative approaches. First, to discriminate the four zones of the Makay's social representation from ICMs, we elaborated a centrality-frequency method that took into account (i) component's frequency of citation, as an indicator of the level of agreement between respondents on each SES component, and (ii) component's median centrality rank, as an indicator of the perceived importance of each component to the SES functioning by respondents (Fig 2, S1 Appendix). The combination of these two metrics was used to classify SES components into the four zones of the representation as defined by [47]: core zone, first periphery, contrasting elements' zone, and second periphery (see S1 Appendix for further details on the four zones). Second, in order to explore the level of ambiguity in individual representations and between groups of respondents, we relied on a quantitative comparison of ICMs based on a set of metrics characterizing ICM structure and content. In particular, we carried out a Principal Component Analysis (PCA) with 9 ICM metrics as active variables (i.e. number of components, number of links, map density, component categories) and with respondent type (i.e. researcher, association member, tour operator, project member) as a supplementary variable (S3 Table in S1 Appendix). The PCA was followed by a Hierarchical Clustering on Principal Components (HCPC) that allowed us to identify respondent clusters based on their representation of the Makay (Fig 2). Both PCA and HCPC were performed with the FactoMineR R package [48]. Third, in order to further explore how ambiguity between people's points of view is very concretely impacting how people understand environmental problems and possible solutions, we focused on two issues related to (i) anthropogenic fires and (ii) provisioning ecosystem services, highlighting how both were differently perceived by different clusters of respondents.

## 4. Results

### 4.1. The social representation of the Makay SES

The 32 respondents we interviewed described the Makay SES through a total of 162 components that we regrouped into 32 types and six categories for the sake of interpretability and classified into the four zones of social representations (Fig 3).

In the core zone of the Makay representation, we found 31 components that were both frequently cited by respondents and with the highest centrality scores (Fig 3; S2 Table). In particular, for most respondents, the Makay was characterized by its unique geomorphology marked by deep canyons created by erosion and crossed by numerous rivers (Fig 4A). As a consequence, the region was considered particularly difficult to access for people, making it a well-preserved area in terms of natural habitats (forests, overall landscapes) and biodiversity (including, in particular, several endemic and flagship species). Notwithstanding this enclosure, local people were acknowledged as central to the Makay SES. Respondents especially emphasized the importance of zebu raising as the main livelihood of local people (Fig 4B), warning about zebu thieves (*dahalo* in Malagasy language) who were considered responsible for an atmosphere of insecurity. In addition, respondents highlighted the intangible connection between local people and their environment that manifest itself through the presence of

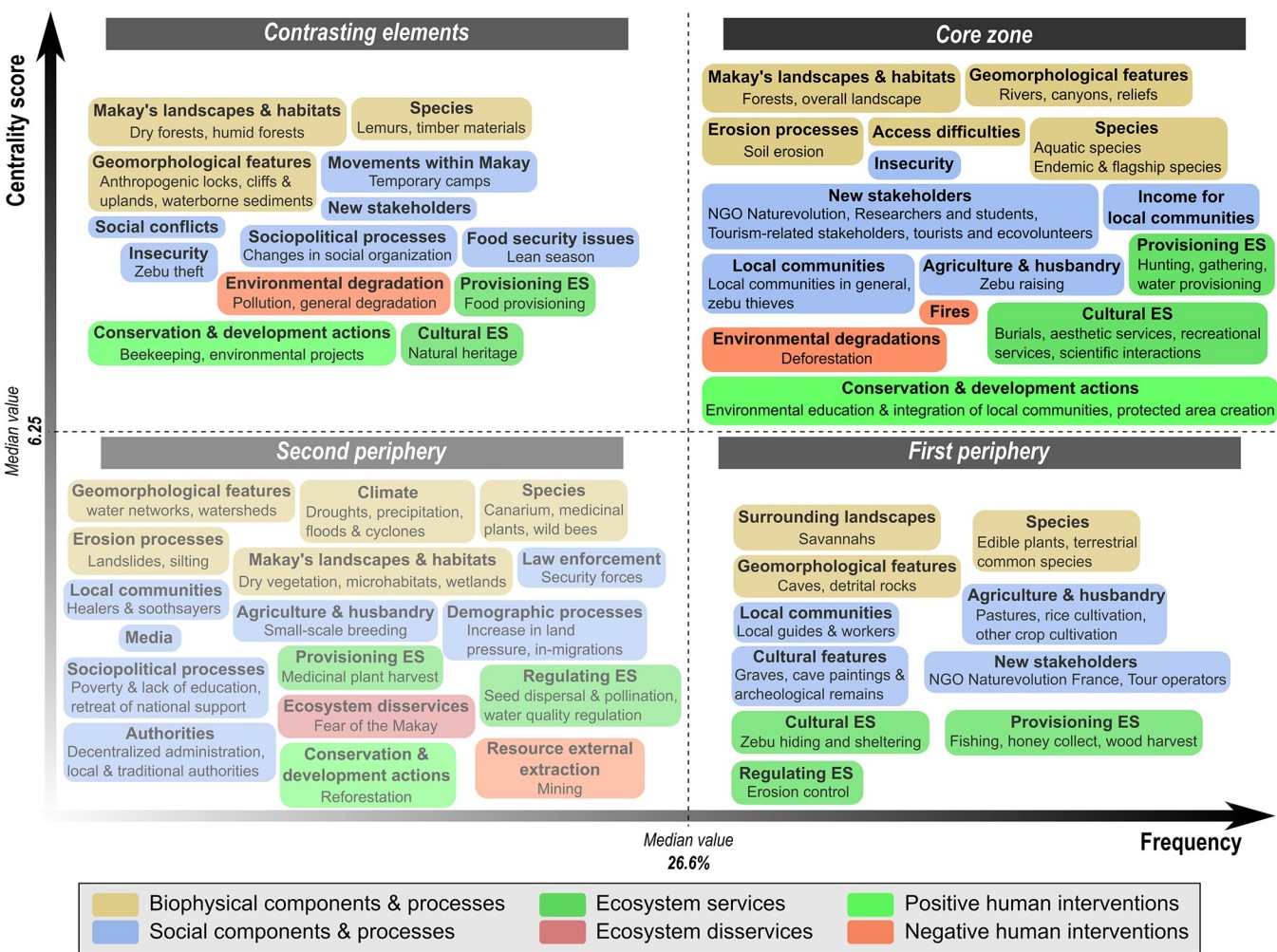

**Fig 3. Content and structure of the Makay social representation held by the respondents.** The four zones of the representation were named following [47], and the classification of elements was done based on centrality-frequency criteria (see S1 Appendix). Boxes correspond to component types, which regroup the components cited by respondents, and the color indicates the category. Only components cited by >10% of the respondents are represented. See S2 Table in S1 Appendix for detail.

burial areas. Yet, according to respondents, local people mostly benefited from the concrete, tangible benefits they derive from the Makay SES, in particular the provisioning services associated with hunting, picking, and water provisioning. On the contrary, most of the intangible benefits and cultural ecosystem services–including scenic value, recreation, and scientific interactions–were seen to be enjoyed by other types of stakeholders such as local NGOs, researchers and students, and tourism-related stakeholders. These 'new', external stakeholders were associated with conservation and development actions, contributing to the creation of the new protected area while offering new sources of income for local people through the activities they develop in the area (e.g. touristic tours, beekeeping). In this global and consensual understanding of the Makay SES, the main threats to nature conservation appeared to be related to local people' activities: hunting was highlighted as a threat to all endemic and flagship species indistinctively, whereas anthropogenic fires used for agricultural and pastoral activities were considered a threat to Makay's landscapes and habitats (Fig 4A).

A total of 20 components were located in the contrasting elements' zone of the Makay representation, which contained infrequently-cited yet central components (Fig 3; S2 Table).

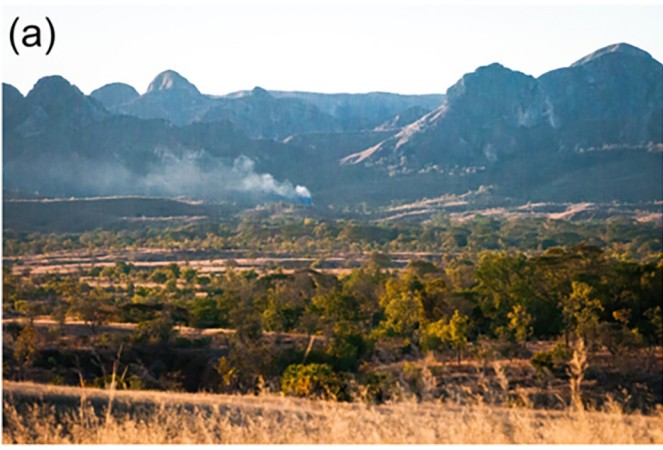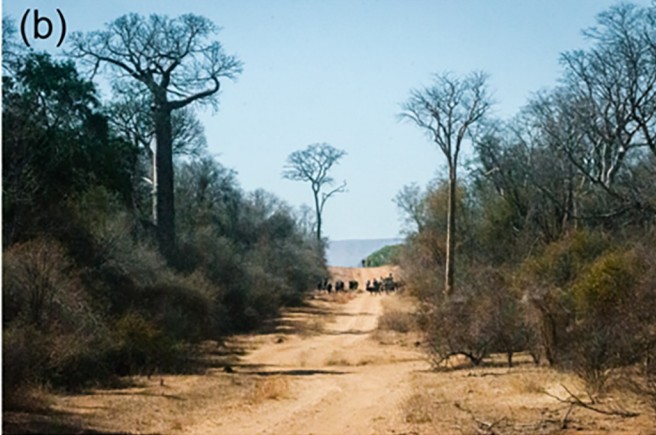

**Fig 4. Photographs of the Makay region (author: S. M. Carrière, date: 2019).** (a) The Makay mountainous massif appears in the background, and an anthropogenic fire is visible at its vicinity. (b) A zebu herd crossing a road.

These components were mainly precisions or generalizations of the central core zone's components. For example, some of them detailed forest types (e.g. dry and wet), specific species (e.g. lemurs), or geomorphological structures (e.g. anthropogenic locks, cliffs and uplands, water-borne sediments). On the contrary, others were more generic and designated 'new stakeholders' or 'cultural ecosystem services' without unpacking these general boxes. As a consequence, these components partly reflected differences in expertise among respondents about the different dimensions of the Makay SES. By corollary, they also reflected components and problems that were considered central but yet unknown by most of the respondents such as human migrations within the Makay region, local social conflicts and sociopolitical issues, or food security problems.

In the first and second peripheries of the representation, we found 49 components that had the lowest centrality scores (Fig 3; S2 Table). They included specific plant species such as edible plants, terrestrial common species, *Canarium* spp. (commercial timber), medicinal plants, and wild bees. Contrary to the species perceived as central, these appeared to contribute to food provisioning but without being threatened by local people. These components also detailed certain aspect of agropastoral activities relative to pasture management, rice cultivation, and small-scale breeding. Interestingly, regulating ecosystem services such as erosion control, seed dispersal and pollination, and water quality regulation were part of the peripheral zones, as well as several provisioning services benefiting to local people relative to fishing, honey harvesting, and medicinal plant gathering. Many social processes were also in the periphery of the representation, including local demographic processes (increasing land pressure, in-migration) and local security forces and authorities (decentralized administrations, local and traditional authorities). Finally, the external extraction of natural resources through mining activities was seen as a peripheral problem.

In sum, in order to provide a synthetic vision of the Makay as highlighted by the 32 respondents, we pooled ICMs into a SCM that allowed to emphasize the Makay's multiple components and their interactions (Fig 5).

## 4.2. Variability in representations at individual and group level

Out of the 162 components cited by all respondents to describe the Makay SES, each respondent mentioned between 20 and 97 components (i.e. between 12% and 60% of all cited components), with an average of 41.5 components ± 15.2 (SD) per respondent. This result testified of

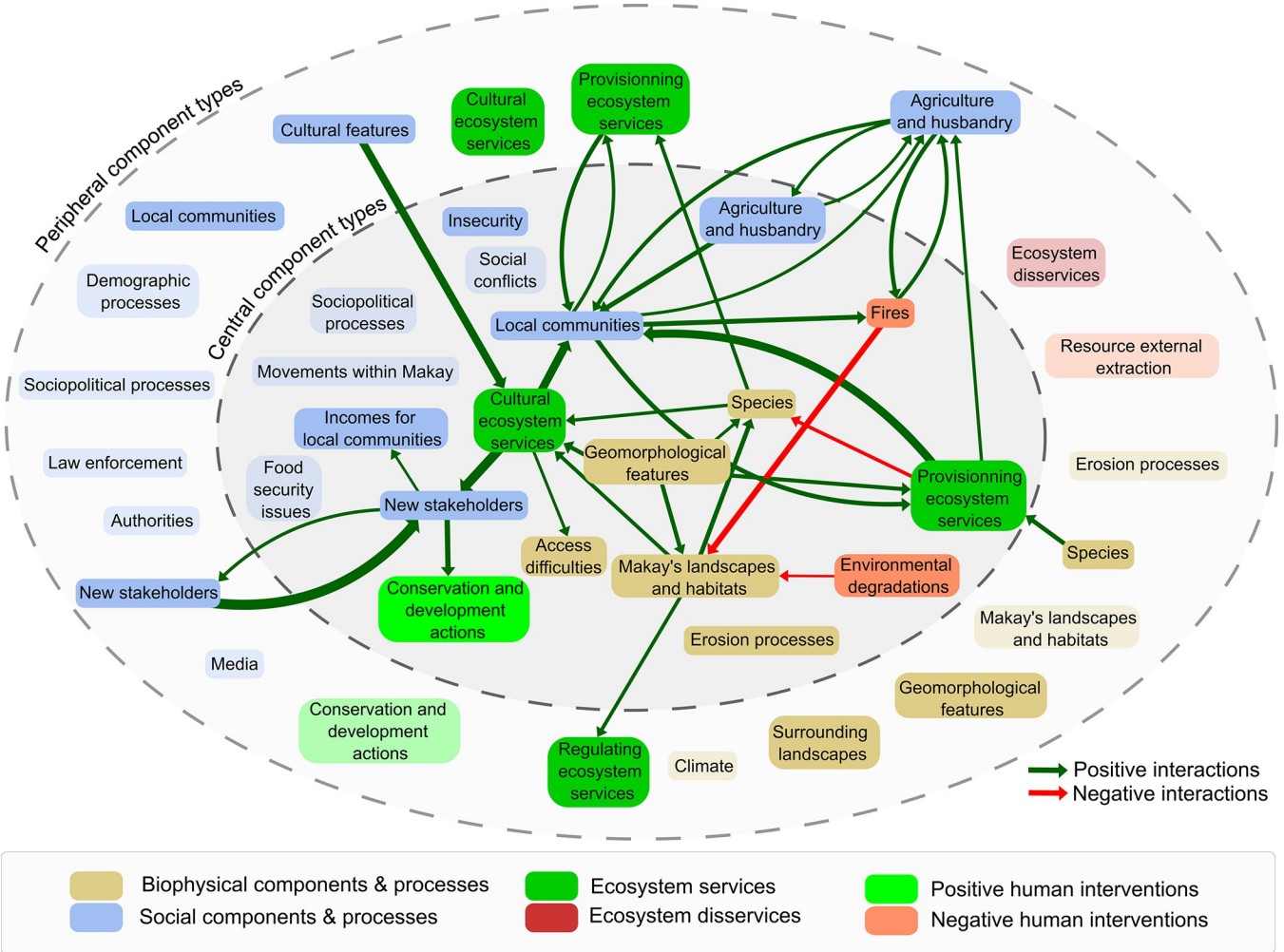

**Fig 5. Social cognitive map of the Makay SES based on the aggregation of the 32 individual cognitive maps and represented at the level of component types for the sake of visibility.** The central component zone contains components with high centrality scores, while the peripheral component zone contains components with low centrality scores (see S1 Appendix). Lightly colored component types contain only components that were cited by less than 26.6% of respondents (median value of the centrality score, see Fig 3). Edges correspond to the positive (in green) and negative (in red) interactions between component types, and their width is proportional to their occurrence in individual cognitive maps (only interactions cited by >10 respondents are represented).

the different visions that the respondents had about the Makay SES, which might either come from differences in expertise or in personal sensibility for certain topics. In order to further understand this heterogeneity, we first quantitatively analyzed variability in individual representations through a PCA and an HCPC based on the structure and content of the ICMs (Fig 6).

The first axis of the PCA (explaining 30.2% of the total variance) was mainly structured by the number of components (i.e. nodes) in an ICM, and by the total number of interactions between components (i.e. edges). The second axis (22.5% of the variance) was mainly structured by the density of the ICMs and by the proportion of ecosystem services. Thus, the two first axes of the PCA explained 52.5% of the total inertia and discriminated ICMs with a large number of components that also contained a large proportion of social components (which varied from 25.0% to 57.9%, with an average of 39.4% ± 8.3) from ICMs with a smaller number of components that gave more room to biophysical components (that represented between 15.8% to 50.0% of ICMs' components, with an average of 29.5% ± 7.4).

## (a) Individuals and clusters

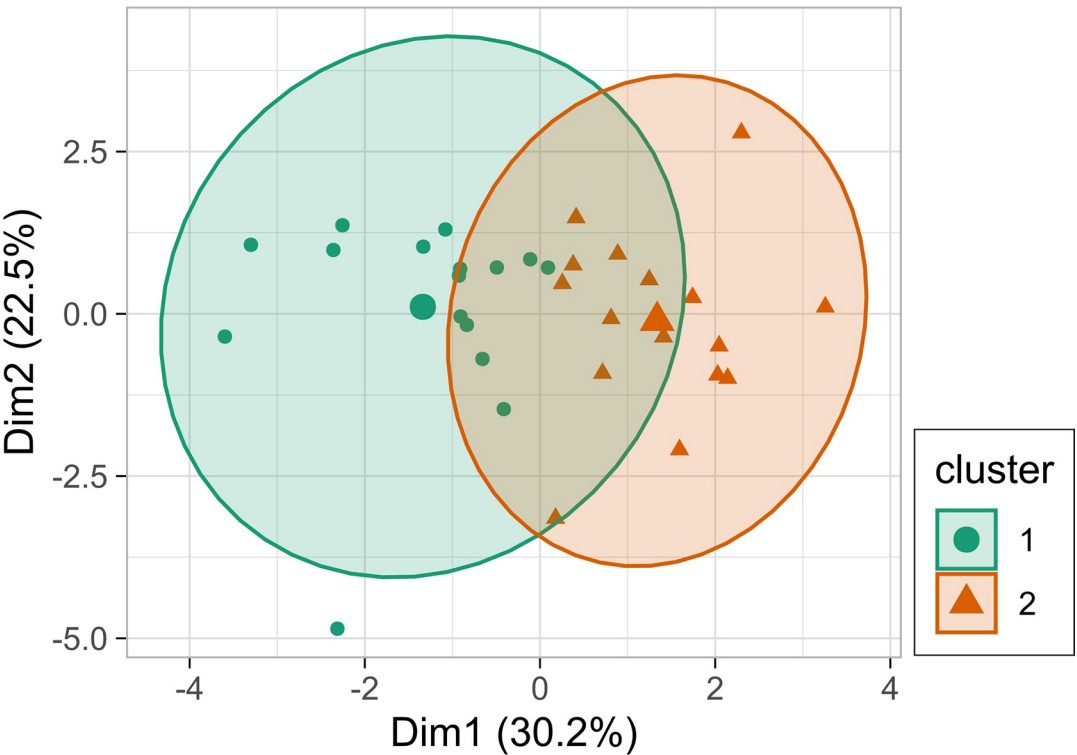

## (b) Active variables

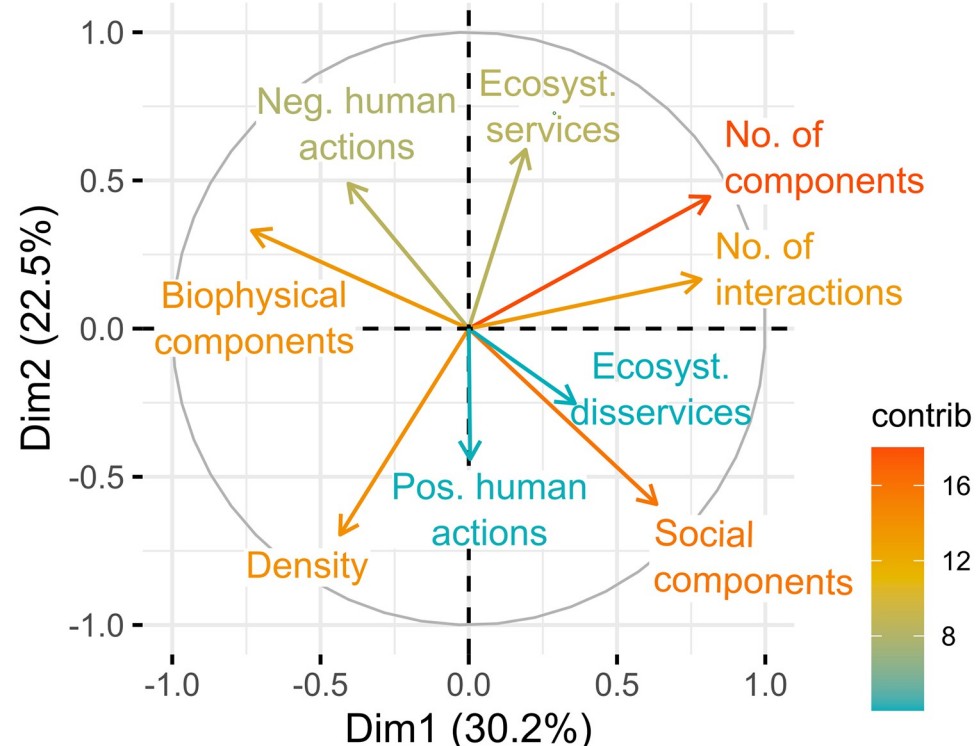

**Fig 6. Variability in individual cognitive maps explored with a Principal Component Analysis (PCA) and Hierarchical Clustering on Principal Components (HCPC).** (a) Projection of the 32 ICMs in the first two axes of the PCA and delineation of the two clusters identified with the HCPC; Cluster 1: ecocentric, Cluster 2: social-ecological. (b) Correlation circle of the quantitative active variables (colored according to their contribution) with the two first axes of the PCA.

In line with this result, two clusters were identified by the HCPC (Fig 6A), highlighting a difference between respondents having an ecocentric vision or a social-ecological vision of the Makay SES (Table 1). On average, respondents in the social-ecological cluster (mainly local NGO members and researchers from the Makay interdisciplinary research program) produced ICMs with more components and interactions than respondents in the ecocentric cluster (mainly composed of the other researchers and tour operators). Furthermore, while giving more emphasis to social components, respondents in the social-ecological cluster reported less negative human actions than respondents in the ecocentric cluster, who gave more emphasis to biophysical ones.

This difference between the ecocentric and the social-ecological visions was reflected in the associated group cognitive maps (S1 Fig), but also in the content and structure of the representation held by the two groups (S2 Fig). In particular, 26 components were common to the two core zones of each group representation, representing 79% and 57% of the core zone elements of the ecocentric cluster's representations and of the social-ecological cluster's representations, respectively. Respondents within the social-ecological cluster gave more emphasis to the plurality of actors of the Makay SES, as well as to local social issues and conflicts (e.g. insecurity, food security problems, and demographic processes). The comparison of the two representations also highlighted differences between the two clusters in their understanding of environmental degradation issues: while respondents in the ecocentric cluster perceived them as resulting from local activities (e.g. trampling and overgrazing, loss of aquatic habitats), respondents in the social-ecological cluster rather pointed to external threats (e.g. mining, illegal trade in wildlife).

**Table 1. Characteristics of the ecocentric vs. social-ecological clusters identified in respondents' individual cognitive maps (ICMs).**

| Cluster characteristics | The ecocentric cluster (N = 16) | The social-ecological cluster (N = 16) | Student test (p-values) |
|---|---|---|---|
| **Stakeholders** | 9 researchers | 3 researchers | |
| | 4 tour operators | 1 tour operator | |
| | 2 local NGO members | 7 local NGO members | |
| | 1 NPA project member* | 5 NPA project members* | |
| **Network metrics** | | | |
| No. of components | 29.1 ± 7.6 | 45.8 ± 10.1 | <0.001*** |
| No. of interactions | 41.8 ± 13.9 | 81.2 ± 36.4 | <0.001*** |
| Density | 0.054 ± 0.018 | 0.040 ± 0.012 | 0.019* |
| **Component categories (in proportion)** | | | |
| Biophysical components | 0.33 ± 0.07 | 0.26 ± 0.06 | 0.009** |
| Social components | 0.36 ± 0.06 | 0.43 ± 0.09 | 0.008** |
| Ecosystem services | 0.21 ± 0.05 | 0.20 ± 0.04 | 0.85 |
| Ecosystem disservices | - | 0.01 ± 0.02 | - |
| Positive human actions | 0.03 ± 0.03 | 0.04 ± 0.03 | 0.51 |
| Negative human actions | 0.08 ± 0.03 | 0.05 ± 0.02 | 0.002** |

The two clusters are characterized by a set of nine descriptors (mean values ± standard deviation), and compared with a Student's t test.

* refers to the interdisciplinary research project evaluating the new protected area

## 4.3. Complementary perspectives about Makay's conservation challenges

To complement quantitative analyses, we undertook a comprehensive analysis focusing on two key conservation challenges that were consensually identified by respondents and the two abovementioned clusters, namely fires and provisioning ecosystem services for local people (S2 Fig). More precisely, fires appeared to be the primary direct perceived threat to Makay's landscapes and habitats while local people, through their subsistence activities (agriculture, husbandry, hunting, gathering) were seen as the main threat to Makay's species. Besides, the issue of fires is well known in Madagascar [49–51], as is the importance of natural resources for the subsistence activities of local people [28]. However, a more detailed look at how these challenges were seen by the two clusters highlighted two contrasting ways of conceiving these challenges and, ultimately, of addressing them.

There was a consensus between respondents about the threat that fires represent to Makay's biodiversity and ecosystems but also on the cause of fires, mainly local people who use fires for their agropastoral activities (Fig 7). However, respondents in the socio-ecological cluster described the causes and consequences of fires with much more details than respondents in the ecocentric cluster. Furthermore, the two clusters had diverging views on (i) fire causes, (ii) the way to regulate them, and (iii) the role of new stakeholders in fire dynamics. While the eco-centric cluster only cited local people activities as a cause of fires, the social-ecological cluster pointed to alternative emerging or increasing social factors that accentuate fire dynamics, such as increasing land pressure as well as changes in the organization of zebus' thieves who use fires as an escape strategy. Respondents in this cluster also considered that the arrival of new actors in the Makay had an impact on fire dynamics: on the one hand, tourists and ecovolunteers might be a cause of unintentional fire outbreaks, and on the other hand, fires may be set by local people to protest against the arrival of new stakeholders. Consistently with these different viewpoints about fires between the two clusters, the fire regulation factors they identified

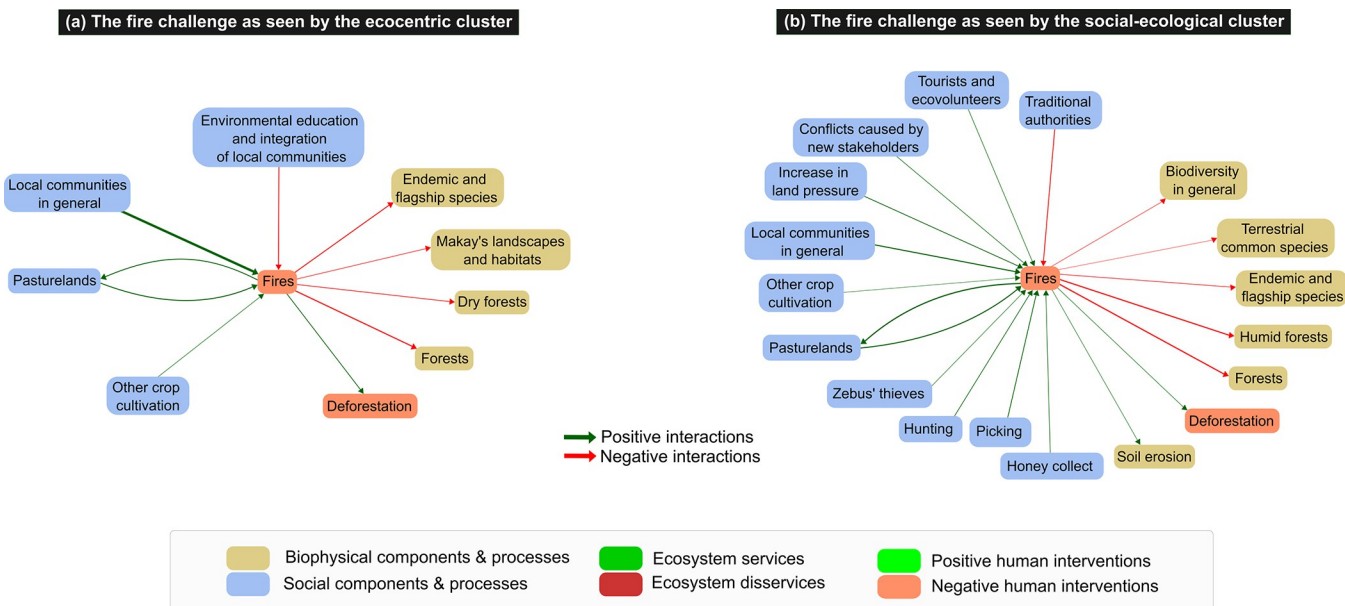

**Fig 7. Group cognitive maps centered on fires in the Makay SES for each of the two clusters of respondents.** Maps are based on the aggregation of the individual cognitive maps for each cluster and represented at the level of components. Only components linked with fires are represented. Edges correspond to the positive (in green) and negative (in red) interactions between components, and their width is proportional to their occurrence in individual cognitive maps (only interactions cited by >2 respondents are represented).

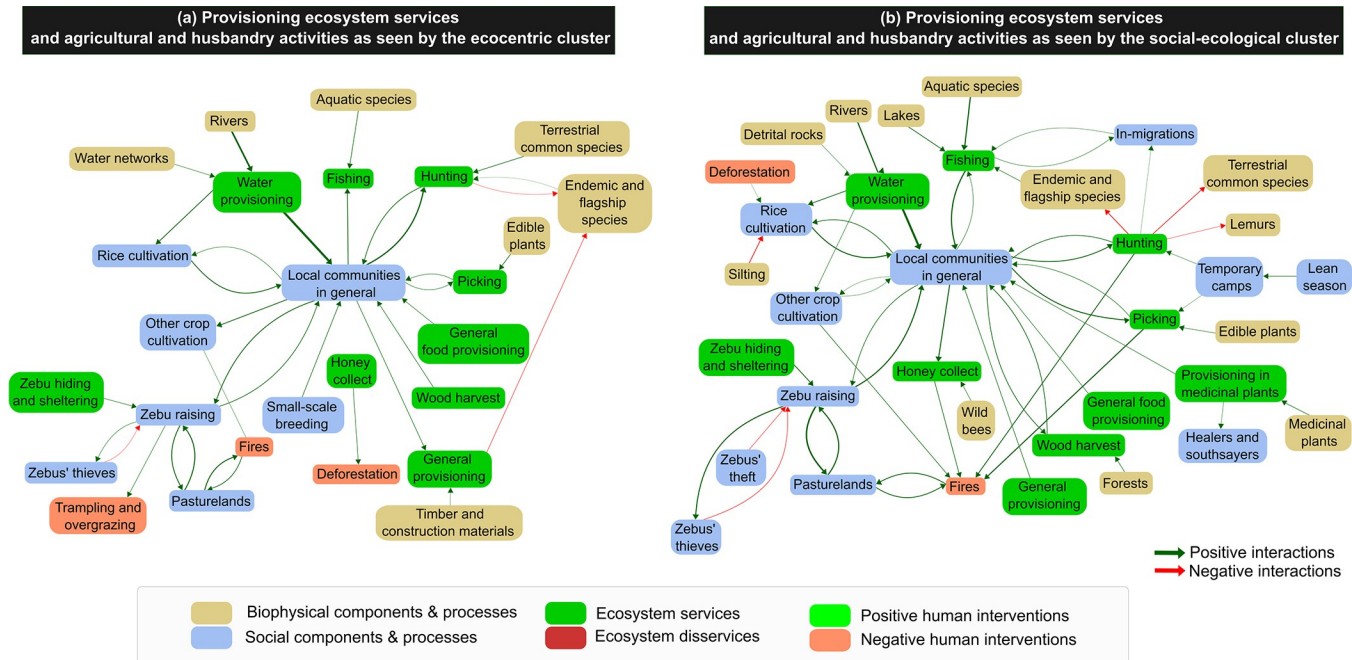

**Fig 8. Group cognitive maps centered on provisioning ecosystem services and agricultural and pastoral activities in the Makay SES for each of the two clusters of respondents.** Maps are based on the aggregation of the individual cognitive maps for each cluster and represented at the level of components. Only components linked with components belonging to the types 'Agriculture and husbandry', 'Food production issues', 'Food security issues' and 'Provisioning ecosystem services' are represented. Edges correspond to the positive (in green) and negative (in red) interactions between component types, and their width is proportional to their occurrence in individual cognitive maps (only interactions cited by >2 respondents are represented).

differ: for the ecocentric cluster, environmental education actions and integration of local people in the PA project should help regulate fires, whereas for the social-ecological cluster, local authorities were the key leverage to act against fires.

Regarding subsistence activities and provisioning ecosystem services management, both clusters emphasized the importance of rice and other crop cultivation activities for local people, as well as the centrality of zebu raising (Fig 8), which were part of the core zone of both clusters' representations (S2 Fig). These activities directly depended on several key ecosystem services, including water supply and the sheltering of zebu herds that both relied on the unique Makay's geomorphology. Many provisioning services were also acknowledged by the two clusters of respondents, such as water provisioning, fishing, hunting, picking and honey collect. For both clusters, hunting was perceived as a direct threat to many Makay's species. Besides these commonalities, the visions of two clusters were yet diverging on several aspects. First, the social-ecological cluster mentioned medicinal plants as key provisioning services for local people, contrary to the ecocentric cluster (S2 Fig). Second, the ecocentric cluster incriminated zebu raising as a cause of fires, trampling and overgrazing, which was not part of the social-ecological cluster's representations (Figs 7 and 8). Third, the social-ecological cluster pointed to threats that concern local people themselves, highlighting for example the negative impact of silting, linked to soil erosion and caused by fire and deforestation, on rice cultivation (Fig 7). Finally, the social-ecological cluster identified several linkages between ecological and social factors to explain current biodiversity challenges. In particular, respondents highlighted the existence of a lean season that local people cope with in establishing temporary camps in the Makay to hunt and pick. This cluster also mentioned people who were not living in the region and come from other parts of the country to hunt in the Makay.

## 5. Discussion

Promoting synergies between nature conservation initiatives and local development remains a key challenge to improve the social-ecological success of terrestrial PAs [51–54]. Inclusive PA management designs that take into consideration the different people's viewpoints about how to address this complex challenge are increasingly advocated as a sound strategy towards this end [55]. In combining mental-model and social representation approaches, we provided in this study a first assessment of the different viewpoints that coexist among some key stakeholders about the Makay region and its on-going social-ecological dynamics. In particular, we showed that this combination of approaches was promising to explore people's multiple perspectives about a given social object or situation, even though some limitations must be pointed. Furthermore, and as in other studies in Madagascar [4, 56], we emphasized two types of representations, one ecocentric and one social-ecological, that might lead to contrasting recommendations for the management of the Makay PA. In highlighting commonalities and divergences in people's views, we argue that this preliminary study offers a unique basis to foster a more inclusive conservation agenda for the Makay region, which we will further discuss below.

### 5.1. Synergies between mental-model and social representation approaches

Several scholars have emphasized that the social representation theory brings valuable insights to the understanding of sustainability and nature conservation problems, either by its own [57–61] or when combined to other theories such as the framing theory [7] or the place theory [62]. While few studies have suggested that combining social representation and mental models could be a valuable attempt [44, 63], our work provides a way to operationalize this combination, allowing us to bring new insights on the understanding of ambiguity issues in the field of conservation biology.

On the one hand, we argue that the social representation theory provides interpretative tools to better understand (i) how an individual may acquire a mental model [6], and (ii) how to deal with ambiguity depending on whether it concerns the core or the peripheral system of a representation. On the other hand, we argue that mental models, in particular through cognitive mapping, provide a systemic understanding of people's representations of an SES that is not emphasized by conventional social representation approaches. Furthermore, graph theory metrics derived from cognitive maps can help take the structural approach of social representations a step further. We will here illustrate these arguments in focusing on some of our key results that revealed (i) the existence of consensual views between different people's representations about the Makay SES and the challenges that the new PA will have to tackle, (ii) the presence of differences in the peripheral system of the Makay representation, and (iii) the presence of differences in the core zone of the representation between two subgroups of respondents.

First, the core zone of the Makay representation highlighted a consensus among respondents on, for instance, the tensions between people livelihoods and nature conservation. In particular, fires and deforestation caused by local people were consensually perceived as a threat to Makay landscapes and species. This consensus highlights that, as elsewhere, reconciling development and conservation goals will be a specific challenge in this region. Yet, as suggested by the two group cognitive maps (Figs 7 and 8), stakeholders were not necessarily on the same page regarding the dynamics associated with these problems, and therefore regarding the solutions to solve them. In this case, we showed that this is the core zone of the representation that differed between the two clusters of respondents (S2 Fig), which could lead to substantial and long-lasting tensions between stakeholders. Indeed, the core zone is the most

resistant and the least negotiable part of a representation, so changing it would imply a complete transformation of the representation and of the prevailing values and beliefs in which the representation is rooted, which is particularly challenging [42]. This first type of ambiguity is therefore supposed to persist in time [64] and may jeopardize actors' capacity to reach a consensus on SES management strategy and conservation practices. Tackling this type of ambiguity that concerns the core zone of the representation might require dialogue and negotiations between stakeholders, and ultimately that each one accepts the legitimacy and validity of values and opinions that they do not necessarily share for the sake of the collective interest.

Second, the identification of the peripheral system of the representation allowed us to point to a second type of ambiguity in people's representations of the Makay. Theoretically speaking, the elements of the peripheral system might not be critical sources of disagreements among actors, as they rather tend to reflect differences in awareness or expertise on specific aspects but not in the substance of the representation itself [42, 64]. For example, in the Makay, the conservation of specific ecosystems (e.g. dry and humid forests) and species (e.g. lemurs), as well as food security issues, were part of the contrasting element zone. This suggests a difference in expertise between respondents, but not necessarily a profound divergence of opinions. Indeed, people's representations might differ on the nature and number of (i) the entities they use to describe reality and/or (ii) the relations they perceive between these entities. Such differences does not necessarily mean that people disagree with each other [58], but simply that they do not have the same level of knowledge or expertise on every aspect of the SES [58, 65]. Yet, cognitive maps also allowed us to point to differences in the way people perceived relations between entities as positive or negative. These differences in people's judgments reflect diverging opinions or representations about a given phenomenon rather than differences in awareness or knowledge. In common social representation research, there is no explicit distinction between how people perceive the entities that compose reality, and how they perceive the interactions between these entities. In allowing to make this distinction, it appeared that cognitive mapping was helpful to distinguish (i) differences in people's awareness, through the diversity and precision of the components they mobilize to describe a reality from (ii) areas of disagreement, through the opinions they have on the interactions between these components. This distinction is particularly important and the two do not have the same consequences for achieving a consensual PA management strategy.

## 5.2. Challenges in combining mental models and social representations

As for all theories and methods, social representation and mental models have well-documented caveats that should be considered prior to their use. Furthermore, we would like to emphasize some key methodological challenges in the combination of the two approaches that are important to consider.

First, the social representation theory has been mainly criticized for its theoretical ambiguity, social determinism focus, cognitive reductionism and lack of a critical agenda [66]. Some of these critics seem to come from a misunderstanding about the concept, as 'social' was sometimes understood as 'shared' whereas in the Moscovici's view it emphasized the fact that a representation is collectively and culturally constructed and validated, but not necessarily shared [67]. Other critics have more serious theoretical and methodological implications, e.g. in pointing to the difficult integration of the complex and dynamic relationship between individual and social agency [66]. Furthermore, specific critics were formulated on the structural approach of representations [64]. Regarding our work, one main limitation is related to the fact that we identified both the content and structure of representations from a single interview with each respondent. While this is a common research practice [43], some authors

recommend to conduct two separate interviews for successively identifying the content of the representation and then its structure [47]. In addition, from the elements that respondents have freely cited, we merged synonyms and similar terms a posteriori, without asking respondents to validate this merging. This implies that, despite the precautions we took during the condensing procedure (S1 Appendix), our own subjectivity on what people said has influenced our results. A post-validation phase might have been useful to confront our choices with respondents opinions [41], which would have required another round of interviews that was not possible for this study.

Second, mental-model approaches, including cognitive mapping, also have a number of practical and theoretical limitations that have been extensively discussed elsewhere [24, 35, 68]. Among other things, mental-model research was criticized as it requires respondents to cognitively engage in the elicitation process, which can vary from one respondent to another. Furthermore, mental models offer a snapshot assessment of continually evolving representations that might not necessarily reflect people's attitudes and behaviors in real-world situations. In particular, the elicitation procedure and context, as well as the interviewee-interviewer relationship, are known to greatly influence the resulting cognitive map [69–71]. As a consequence, the ICMs we obtained in our study did not necessarily reflect how respondents frame the Makay SES in their own internal mental model. Rather, an ICM should be understood as the result of a socially situated interaction where both the researcher and respondent had an active role [72]. Finally, we note that cognitive maps are (over)simplified models of complex social objects, and our analytical procedure led to a further purposeful simplification. While the diversity and depth of thought provide useful information [71], cognitive map simplification was necessary to perform quantitative analyses but also to avoid the Bonini paradox: a model closely representing the complexity of a real system might become incomprehensible and not transparent [73]. Our approach therefore resulted from a trade-off between embracing complexity and producing intelligible and comparable maps, which was mandatory to articulate qualitative and quantitative analyses.

Finally, our ambition to combine mental models and social representations led us to make certain methodological choices, in particular regarding how we investigated the content and structure of Makay representations. There are effectively different methods to study these, each one having its specific caveats [74], and in particular rank-frequency and importance-frequency criteria that are widely used to distinguish the four zone of a social representation [43]. In our work, we devised a new method based on centrality-frequency criteria. The use of a centrality measure has already been used in social representation research to assess the 'qualitative centrality' of the elements that belong to the core zone of a representation, whereas frequency reflects their 'quantitative centrality' [75]. Our method therefore is an adaptation of existing methods that put graph theory metrics to contribution, resting on the assumption formulated in cognitive-map research that nodes with higher centrality are perceived by respondents as more important to the system [76]. The centrality-frequency method has not less weaknesses than existing methods, but it allowed to operationalize the bridge between social representations and cognitive maps. A comparison between the different methods could be insightful to further discuss their respective strengths and weaknesses as well as epistemic implications.

## 5.3. Recommendations for the Makay's protected area

This mixed, in-depth analysis of the Makay's representations among key stakeholders allowed us to draw some key lessons for the future management of the PA in the perspective of a more just and sustainable nature conservation initiative. However, these lessons should be taken with caution as our study did not include, at this stage, the viewpoints of local people living

around the Makay massif. Integrating their visions to our analysis is a key research perspective that will require an adaptation of our method. Indeed, the elicitation procedure we used (i) relied on the Mental Modeler software, (ii) was adapted to remote interviews, and (iii) was involving a graphic representation of the Makay SES. As a consequence, this procedure would likely not to be suited to collect data on the representations of the local people who live in the Makay region who are not familiar with computers and graphics, and who will certainly be more comfortable with more indirect elicitation methods (for example, see the method proposed in [77]).

First, achieving a more precise understanding of local people and livelihoods, and how they interact with their environment, seems a preliminary mandatory objective for formulating relevant recommendations for the Makay PA management strategy. Indeed, respondents commonly acknowledged the importance of local people in the ecological dynamics of the Makay, in particular through their subsistence activities that might sometimes be a threat to the Makay's ecosystems and unique biodiversity. However, in the absence of context-specific studies, it seemed that respondents' visions were mainly based on preformatted opinions based on their prior experiences in other regions or readings. The use of fire by local people and the impact on the Makay ecosystems, generally highlighted by the respondents, illustrates this argument and pleads for fostering local research on this issue, which is a key challenge in the whole Madagascar's island but with very contrasting local realities [78]. Similarly, the actual impact of zebu grazing, as mentioned by the ecocentric cluster, should be precisely documented.

Second, promoting dialogue and social learning between stakeholders should be another central objective for PA managers. Indeed, our results revealed two clusters of respondents holding different representations of the Makay, which should be considered to make them complementary, to prevent conflicts and to take into account all the dynamics and issues of the Makay region. While the ecocentric cluster only highlights the responsibility of local populations and their activities in the degradation of Makay's ecosystems, the social-ecological cluster emphasizes the existence of other causes of fires, places hunting and picking activities in the local socioeconomic context, and also mentions other potential threats. The former testified to a vision that may lead to advocating for 'a back to the barriers' conservation strategy [79], while the latter acknowledged that conservation and development issues are inseparable, advocating for a more inclusive, co-constructed conservation strategy [2]. Reconciliation might therefore represent a challenging and long-lasting process in Madagascar, which could only be fostered by a nourished dialogue between stakeholders, fed by the new scientific evidences that we advocated as necessary in our previous point (also see [32]). Furthermore, beyond commonly-acknowledged challenges, researchers and PA managers must not neglect more subtle dynamics and emerging issues mentioned by the social-ecological cluster. In particular, our results pointed to mining activities and wildlife trade as potential threats to the Makay's ecosystems and biodiversity, even though a limited number of respondents were aware of these. Considering that mining activities could rapidly expand and pose major environmental degradations [80], such weak signals should not be overlooked.

Finally, it seems crucial to ensure that the creation and management of the PA is integrated into local power dynamics. Indeed, some respondents mentioned the place of traditional local authorities, decentralized administrations and national authorities in the Makay SES, as well as conflicts between stakeholders. On the one hand, our results tend to point to a disconnection between the local dynamics of the Makay region and the national authorities, with respondents mentioning the poverty of local people, difficult access to education, and a general retreat of national support in the region. A better understanding of the interactions between the population, local power holders (formal and informal) and national authorities is all the more

essential as the weakness of intermediary bodies to link elites and ordinary citizens has been identified as one of the main causes of Madagascar's long-term instability and decline [81]. On the other hand, since the local authorities are rarely mentioned by the respondents and have little connection with the dynamics of the creation of the PA, it appears that they have little involvement in the project. However, a real involvement of local authorities in the creation and management of the PA is essential for the participation and ownership of the PA by local people and for the PA sustainability [82].

## 6. Conclusion

This study presents a first assessment of different stakeholders' representations about the Makay region and the key socio-environmental challenges that the newly-created PA will have to tackle. In combining mental models and social representation theory, it revealed both consensual and non-consensual elements in people's representations, and discriminates two visions of the Makay. The first vision was ecocentric, and insisted on the threat of local factors and local people on biodiversity, such as anthropogenic fires, hunting, and deforestation. The second vision was social-ecological, and was more complex, also pointing to more external threats such as the conflicts generated by the arrival of new stakeholders (tourists, associations) and mining activities. Even though both visions converged regarding the challenging aspect of concealing local people activities and nature conservation in the region, they yet differed on how to best achieve this conciliation. In order to tackle this ambiguity between stakeholders' viewpoints so it would not undermine the success of the newly-created PA, we recommend an increased research effort in the Makay, as well as the creation of dialogue arenas within the PA governance to foster social learning between stakeholders. Increasing dialogue between local managers and governmental authorities would be another key priority, as well as the integration of local people' knowledge and viewpoint in future research in the region.

## Supporting information

**S1 Checklist. Questionnaire on inclusivity in global research.**
(DOCX)

**S1 Table. List of the 10 references indexed in Scopus and dealing with the Makay region in Madagascar.**
(DOCX)

**S2 Table. Number of components and mean frequency for each component's types, in the four zones of the Makay's social representations.**
(DOCX)

**S1 Appendix. (Including S1 and S2 Figs and S2 and S3 Tables): Methods.**
(DOCX)

**S2 Appendix. Interview guide.**
(DOCX)

**S1 Fig. Group cognitive maps of the Makay SES according to respondents' distribution in the two clusters.**
(DOCX)

**S2 Fig.** Graphs showing the elements of the four zones of the Makay social representations for (a) the ecocentric cluster, and (b) the social-ecological cluster.
(DOCX)

## Acknowledgments

We thank the research laboratories SENS (UMR Knowledge, Environment and Societies), PALOC (UMR Local Heritage, Environment and Globalization), DIAL (UMR LEDa, DIAL) and Art-Dev (UMR Actors, Resources and Territories in Development) for their support. We would also like to thank all the stakeholders who accepted to be interviewed, and in particular Bernard Forgeau for its valuable help throughout this study.

## Author Contributions

**Conceptualization:** Christian Culas, Emmanuel Pannier, Mireille Razafindrakoto, François Roubaud, Stéphanie M. Carrière.

**Data curation:** Céline Fromont.

**Formal analysis:** Céline Fromont.

**Funding acquisition:** Christian Culas, Stéphanie M. Carrière.

**Investigation:** Céline Fromont.

**Methodology:** Céline Fromont, Julien Blanco, Stéphanie M. Carrière.

**Supervision:** Stéphanie M. Carrière.

**Writing – original draft:** Céline Fromont.

**Writing – review & editing:** Julien Blanco, Christian Culas, Emmanuel Pannier, Mireille Razafindrakoto, François Roubaud, Stéphanie M. Carrière.

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
