## [Decision Letter · Decision Letter 0]

6 Apr 2022

PONE-D-22-03075Towards an inclusive nature conservation initiative: preliminary assessment of stakeholders’ representations about the Makay region, MadagascarPLOS ONE

Dear Dr. Fromont,

Thank you for submitting your manuscript to PLOS ONE. After careful consideration, we feel that it has merit but does not fully meet PLOS ONE’s publication criteria as it currently stands. Therefore, we invite you to submit a revised version of the manuscript that addresses the points raised during the review process.

We look forward to receiving your revised manuscript.

Kind regards,

Rodrigo Pereira Medeiros

Academic Editor

PLOS ONE

Journal Requirements:

(This research was funded by the MSH-Sud (Maison des Sciences de l’Homme) through the Makay-Mada project, and by the research laboratories SENS (UMR Knowledge, Environment and Societies), PALOC (UMR Local Heritage, Environment and Globalization), DIAL (UMR LEDa, DIAL) and Art-Dev (UMR Actors, Resources and Territories in Development). We would also like to thank all the stakeholders who accepted to be interviewed, and in particular Bernard Forgeau for its valuable help throughout this study.)

(SC and CC received funding from the "Fondation Maison des Sciences de l'Homme" for this study. The funders had no role in study design, data collection and analysis, decision to publish, or preparation of the manuscript.)

5. We note that Figure 1 in your submission contain map images which may be copyrighted. All PLOS content is published under the Creative Commons Attribution License (CC BY 4.0), which means that the manuscript, images, and Supporting Information files will be freely available online, and any third party is permitted to access, download, copy, distribute, and use these materials in any way, even commercially, with proper attribution. For these reasons, we cannot publish previously copyrighted maps or satellite images created using proprietary data, such as Google software (Google Maps, Street View, and Earth). For more information, see our copyright guidelines: http://journals.plos.org/plosone/s/licenses-and-copyright.

6. We note that Figure 5 in your submission contain copyrighted images. All PLOS content is published under the Creative Commons Attribution License (CC BY 4.0), which means that the manuscript, images, and Supporting Information files will be freely available online, and any third party is permitted to access, download, copy, distribute, and use these materials in any way, even commercially, with proper attribution. For more information, see our copyright guidelines: http://journals.plos.org/plosone/s/licenses-and-copyright.

a. You may seek permission from the original copyright holder of Figure 5 to publish the content specifically under the CC BY 4.0 license. 

Additional Editor Comments:

I congratulate the authors for the manuscript, that, if suggestions are taken carefully, is suitable for publication Please follow the suggestions by reviewer 1, especially on methods, that are crucial for being considered for publication. The manuscript is relevant and innovative, and the authors should review carefully the suggestion to increase the potential of the manuscript for a bigger audience. Also, provide the best resolution possible for your figures in the final submission. Despite being considered a minor revision, I considered it a major revision since the comments are precise to say that the methods need a substantial revision, with the need to adjust the following sections.

Reviewers' comments:

Reviewer's Responses to Questions

**Comments to the Author**

1. Is the manuscript technically sound, and do the data support the conclusions?

Reviewer #1: Partly

Reviewer #2: Yes

2. Has the statistical analysis been performed appropriately and rigorously? 

Reviewer #1: Yes

Reviewer #2: Yes

3. Have the authors made all data underlying the findings in their manuscript fully available?

Reviewer #1: Yes

Reviewer #2: Yes

4. Is the manuscript presented in an intelligible fashion and written in standard English?

Reviewer #1: Yes

Reviewer #2: Yes

5. Review Comments to the Author

Reviewer #1: The paper fits well with the aims of the journal and the targeted audience, providing a very interesting (and useful) background on the need to protect a vulnerable area in Madagascar, while considering the added-value of social representation to ensure the viability of area conservation. The introduction provides enough background on protected areas and the need to be analysed considering SES, but also calls for action. However, Section 2, on Material and Methods, needs to be reshaped. Firstly, I suggest to exclude the "presentation of the Malay region" as a new section clearly focused on the case study, and providing more details on the challenges and risks if the area is not protected. Following, I suggest to organize the rest of the sub-divisions in just two main sub-sections: Data collection and Data analysis. Furthermore, I think that sub-section 2.3 on Ethical approval should not be included in this section in the current detail, as the journal asked for ethical issues when submitting and as a separate issue. Regarding data collection, I was wondering if 32 respondents are a validated sample to ensure that all confronted perspectives are consulted. For example, you specified their profile as researchers, members of NGOs (civil society) and tour operators... but what about managers? and public administration or government? They have no relevance in the decision-making process of the PA creation? The reasons to identify and select respondents must be clearly extended. Likewise, are interviews recorded? How about the transcription process? Does the speeches being codified or organized in some way to highlight shared views among respondents? In p. 18, line 245 it is presented a double focus on anthropogenic fires and provisioning ecoservices, but why both issues are considered to be focused in? The results section seems to identify both issues, so they are a result or a previous selection criteria to analyse data from the interviews?

The results section are clear. I just suggest to edit the title of sub-section 3.2 currently focus on ambiguity but to cognitive maps as the content is focused on that. Sub-section 3.3 seems a social-learning process so maybe the title "different perspectives" is so general... In the discussion, sub-section 4.3 provides recommendations more than implications. The conclusions are focused on and reflect the obtained results.

Reviewer #2: The manuscript is very well written and presented. It tackles a central issue in biodiversity conservation: how to deal with ambiguity and competing perceptions and views about the social and ecological systems submitted to conservation actions. To my knowledge, the authors present an innovative method combining the analysis of mental models and social representation, providing results and insights that form a basis for further research and for the improvement of conservation strategies in the Makay region. One of the main caveats, the fact that local people were not included in the study, is recognized and discussed. Results are presented with the help of well thought out figures and are adequately interpreted and discussed. my recommendation is to accept this manuscript for publication.

6. PLOS authors have the option to publish the peer review history of their article (what does this mean?). If published, this will include your full peer review and any attached files.

Reviewer #1: **Yes: **Sandra Ricart

Reviewer #2: No

---

## [Author Response · Author response to Decision Letter 0]

12 May 2022

Dear Dr. Rodrigo Pereira Medeiros,

First and foremost, we would like to thank you for giving us the opportunity to submit a revised draft of our manuscript entitled " Towards an inclusive nature conservation initiative: preliminary assessment of stakeholders’ representations about the Makay region, Madagascar" to PLOS ONE. We appreciate the time and effort that you and the reviewers have dedicated to providing your insightful and constructive comments on our manuscript.

We have made changes to incorporate the suggestions provided by you and the reviewers. We have highlighted the changes within the manuscript in the attached document called "Revised Manuscript with Track Changes", and have detailed in a point-by-point our responses to the editor’s and reviewers’ comments and concerns. Finally, due to the substantial modification in the bibliography, we also revised the corresponding number of each reference cited in the paper.

We look forward to hearing from you regarding our submission and to responding to

any further questions and comments you may have.

Sincerely,

Céline Fromont, April 28th 2022

Journal Requirements:

1. PLOS ONE's style requirements: We ensured that our manuscript meets PLOS ONE's style requirements.

2. PLOS’ questionnaire on inclusivity in global research: We uploaded a completed version of our questionnaire as Supporting Information. Besides, we added the sentence “Additional information regarding the ethical, cultural, and scientific considerations specific to inclusivity in global research is included in the Supporting Information (S1 Checklist)” in a subsection ‘Inclusivity in global research’ to the Methods section.

3. Acknowledgements and Funding Statement: Thank you for your suggestion. As required, we removed the funding-related text from the acknowledgements, as follows:

“We thank the research laboratories SENS (UMR Knowledge, Environment and Societies), PALOC (UMR Local Heritage, Environment and Globalization), DIAL (UMR LEDa, DIAL) and Art-Dev (UMR Actors, Resources and Territories in Development) for their support. We would also like to thank all the stakeholders who accepted to be interviewed, and in particular Bernard Forgeau for its valuable help throughout this study.”

We would like to update the Funding Statement as follows:

“SMC and CC received funding from the "Fondation Maison des Sciences de l'Homme" for the MakayMada project of which this study is part. The funders had no role in study design, data collection and analysis, decision to publish, or preparation of the manuscript”

4. Data Availability statement: Thank you for your suggestion, we do not wish to make changes to our Data Availability statement.

5. Figure 1: Thank you for being concerned by Figure 1. However, this figure does not use any copyrighted material. The map was built by Auréa Pottier (engineer at IRD), under QGIS, using OpenStreetMap data free to use under an open license. From these data, we made all the layers present on this map. We did not use any other sources for this map.

Accordingly, we made the changes in the caption as follows:

“Figure 1: Location of the Makay region, Madagascar. The map was built by Auréa Pottier (engineer at IRD), under QGIS, using OpenStreetMap data.”

6. Figure 5: Thank you for being concerned by Figure 5. However, this figure does not use any copyrighted material. It was built by the authors with R and Inkscape softwares, from our data. We did not use any other sources for this figure.

Comments from Reviewer #1:

Comment Rev1 #1: I suggest to exclude the "presentation of the Malay region" as a new section clearly focused on the case study, and providing more details on the challenges and risks if the area is not protected.

Response Rev1 #1: We agree with this comment and created a new section “Study site” (line 89) that provides more information about the Makay region. This section contains two subsections “Presentation of the Makay region” (line 90) and “Challenges and potential threats” (line 114) to give more information about the local and national context and potential risks if the area is not protected.

Comment Rev1 #2: I suggest to organize the rest of the sub-divisions in just two main sub-sections: Data collection and Data analysis.

Response Rev1 #2: We thank reviewer 1 for his suggestion. We have renamed the subsection “Elicitation of individual representations of the Makay” to “Data collection” (line 188), and merged the subsections “Condensing and aggregating individual cognitive maps” and “Analysis of social representations and ambiguity in individuals’ views” into “Data analysis” (line 240). Before these subsections, we have kept the "conceptual background" part, because it seems to us necessary as a preamble to explain the articulation of the mental-model theory and the social representation theory.

Comment Rev1 #3: Furthermore, I think that sub-section 2.3 on Ethical approval should not be included in this section in the current detail, as the journal asked for ethical issues when submitting and as a separate issue.

Response Rev1 #3: We followed PLOS ONE requirements and renamed this subsection “Inclusivity in global research” into the “Material and methods” section. We uploaded a completed version of our questionnaire on inclusivity in global research as Supporting Information (S1 Checklist).

Comment Rev1 #4: Regarding data collection, I was wondering if 32 respondents are a validated sample to ensure that all confronted perspectives are consulted. For example, you specified their profile as researchers, members of NGOs (civil society) and tour operators... but what about managers? and public administration or government? They have no relevance in the decision-making process of the PA creation? The reasons to identify and select respondents must be clearly extended. Likewise, are interviews recorded? How about the transcription process? Does the speeches being codified or organized in some way to highlight shared views among respondents?

Response Rev1 #4: We thank Reviewer 1 for pointing this

out. In order to decide when to stop interviewing more respondents, we applied a data saturation technique that is commonly used in social investigations: we considered that enough respondents were interviewed when no new information (new components in cognitive maps) was added by an interview. We explain this data saturation in S1 Appendix (S1 Figure). Regarding the inclusion of managers, we included members of the NGO carrying the project of creation of the protected area and expected to be the future manager of this protected area among the members of local environmental organizations we interviewed. We have added this precision to line 199-200. Regarding public administration and government, we chose to elicit cognitive maps from people who had personally been in the Makay (we have added this precision to line 194), and public administration and government did not have sufficient on-the-ground knowledge of the region to produce a cognitive map of the Makay. However, we did meet with the Ministry of the Environment's Protected Area Creation and Management Department for additional interviews. The interviews were recorded but not transcribed in full, as our main material for data analysis was the cognitive maps produced by the respondents. We used an a posteriori standardization of all cited components in the cognitive maps to allow for analysis and comparisons between respondents. We used the recording of the interviews in order not to betray respondents’ discourses and opinions. We give more information about this standardization procedure in the S1 Appendix.

Comment Rev1 #5: In p. 18, line 245 it is presented a double focus on anthropogenic fires and provisioning ecoservices, but why both issues are considered to be focused in? The results section seems to identify both issues, so they are a result or a previous selection criteria to analyse data from the interviews?

Response Rev1 #5: We thank Reviewer1 for raising this important point. We chose these two issues firstly because they were mentioned by most of the respondents and linked to many components of the Makay system, but also because these two issues are central in Madagascar. We added some clarification in lines 414-416: “Besides, the problem of fires is well known in Madagascar [49–51], as is the importance of natural resources for the subsistence activities of local people [28].”, with the references “49. Kull CA. Madagascar aflame: landscape burning as peasant protest, resistance, or a resource management tool? Political Geography. 2002;21(7):927‑53.”, “50. Frappier-Brinton T, Lehman SM. The burning island: Spatiotemporal patterns of fire occurrence in Madagascar. PLOS ONE [Internet]. 2022;17(3)” and “51. Kull CA. Deforestation, erosion, and fire: degradation myths in the environmental history of Madagascar. Environment and History. 2000;6(4):423‑50.”

Comment Rev1 #6: I just suggest to edit the title of sub-section 3.2 currently focus on ambiguity but to cognitive maps as the content is focused on that. Sub-section 3.3 seems a social-learning process so maybe the title "different perspectives" is so general...

Response Rev1 #6: We changed the title of these subsections to “Variability in representations at individual and group level” and to “Complementary perspectives about Makay’s conservation challenges”.

Comment Rev1 #7: In the discussion, sub-section 4.3 provides recommendations more than implications.

Response Rev1 #7: We agree with Reviewer 1, and changed the title to “Recommendations for the Makay’s protected area”.

Comments from Reviewer #2:

We would like to thank Reviewer 2 for the time spent reviewing our manuscript and for his very positive comments on this work.

---

## [Decision Letter · Decision Letter 1]

15 Jul 2022

Towards an inclusive nature conservation initiative: preliminary assessment of stakeholders’ representations about the Makay region, Madagascar

PONE-D-22-03075R1

Dear Dr. Fromont,

We’re pleased to inform you that your manuscript has been judged scientifically suitable for publication and will be formally accepted for publication once it meets all outstanding technical requirements.

Kind regards,

Randeep Singh

Academic Editor

PLOS ONE

Additional Editor Comments (optional):

Reviewers' comments:

Reviewer's Responses to Questions

**Comments to the Author**

1. If the authors have adequately addressed your comments raised in a previous round of review and you feel that this manuscript is now acceptable for publication, you may indicate that here to bypass the “Comments to the Author” section, enter your conflict of interest statement in the “Confidential to Editor” section, and submit your "Accept" recommendation.

Reviewer #1: All comments have been addressed

2. Is the manuscript technically sound, and do the data support the conclusions?

Reviewer #1: Yes

3. Has the statistical analysis been performed appropriately and rigorously? 

Reviewer #1: Yes

4. Have the authors made all data underlying the findings in their manuscript fully available?

Reviewer #1: Yes

5. Is the manuscript presented in an intelligible fashion and written in standard English?

Reviewer #1: Yes

6. Review Comments to the Author

Reviewer #1: Thank you for considering my suggestions and providing a very extensive answer to each point. Congratulations for a very interesting work.

7. PLOS authors have the option to publish the peer review history of their article (what does this mean?). If published, this will include your full peer review and any attached files.

Reviewer #1: **Yes: **Sandra Ricart

---

## [Editor Report · Acceptance letter]

5 Aug 2022

PONE-D-22-03075R1 

Towards an inclusive nature conservation initiative: preliminary assessment of stakeholders’ representations about the Makay region, Madagascar 

Dear Dr. Fromont:

I'm pleased to inform you that your manuscript has been deemed suitable for publication in PLOS ONE. Congratulations! Your manuscript is now with our production department. 

Kind regards, 

on behalf of

Dr. Randeep Singh 

Academic Editor

PLOS ONE